# The Effect of Magneto-Priming on the Physiological Quality of Soybean Seeds

**DOI:** 10.3390/plants12071477

**Published:** 2023-03-28

**Authors:** Rute Q. de Faria, Amanda R. P. dos Santos, Thiago B. Batista, Yvan Gariepy, Edvaldo A. A. da Silva, Maria M. P. Sartori, Vijaya Raghavan

**Affiliations:** 1Instituto Federal de Educação Ciência e Tecnologia Goiano, Department of Agricultural Engineering, Rod. Geraldo Silva Nascimento, Campus Urutaí, Km-2,5—Zona Rural, Urutaí 75790-000, GO, Brazil; 2Department of Agricultural Engineering, Campus Alegrete, Universidade Federal do Pampa, Av. Tiarajú 810, Alegrete 97546-550, RS, Brazil; 3Department of Production and Plant Breeding, School of Agriculture, Sao Paulo State University (UNESP), Botucatu. Av. Universitária, n° 3780—Altos do Paraíso, Botucatu 18610-034, SP, Brazil; 4Macdonald Campus, McGill University, 21111, Lakeshore Road, Sainte-Anne-De-Bellevue, QC H9X 3V9, Canada

**Keywords:** electromagnetic bio stimulation, seed survival, P50, ultra-high frequency, wavelength, drying modelling

## Abstract

Microwaves have been applied to the drying of seeds of several species due to their maintenance of the quality of the seeds and reduction of time and costs. However, few is known about the effect of microwaves on the increase of the physiological quality of soybean seeds and especially their effects on longevity. Therefore, the use of microwaves as magneto-priming in soybean seeds was the object of study in this work. For this purpose, two soybean cultivars were selected and submitted to the ultra-high frequency (UHF) microwave exposure of 2.45 GHz, in the wavelength of 11 cm, and power of 0.2 W/g, for 15 min. The results showed that this condition of exposure to the microwave brought benefits in both cultivars after treatment. Incremental improvements were observed in the germinability indexes, the seedling length, the water absorption by the seeds, the fresh mass, dry mass, and longevity. The genes related to seed germination and longevity showed superior expression (HSFA3, HSP21, HSP17.6b, EXP, ABI3) with magneto-priming treatment. The data found ensure the use of the technique as a viable option for pre-treatment as magneto-priming in soybean seeds in order to improve seed quality.

## 1. Introduction

The use of electromagnetic energy to increase the performance of seed germination and vigor has been shown to be an option of low ecological impact, with the possibility to be used on a large scale [1]. The application of microwaves in the drying of seeds without prejudice to germination has been the object of research in several studies, which defined safe conditions for drying soybean seeds with the use of microwaves in the frequency of 2.45 GHz [2].

Other studies have reported the bioeffect and stimulation in seeds, verified in rice seeds, a significant increase in germination indexes, and in physiologic quality. The authors identified an increase in the germination indexes of wheat seeds, chickpeas, and green mung beans when submitted to a low-powered microwave field [3,4].

The microwaves used in this technique in wheat seeds not only act as a bio-stimulus, but also had a prophylactic effect against pathological microorganisms [5]. In soybean seeds, the use of electromagnetic energy outside the microwave spectrum improved the germinability indexes of seeds, water absorption, seedling length, and fresh and dry mass, and reduced the action of antioxidant enzymes [6,7,8,9].

The first studies reporting the benefits of using electromagnetic energy in seeds date back more than 60 years. In [10], the author, identified that carrot and onion seeds increased germination percentages when exposed to radio frequency waves. The studies also found that specific conditions of microwave exposure during drying allowed preservation of the content of tocopherols in soybean seeds [11]. The impressive improvement in microwave technology has shown the possibility to use microwaves in drying seeds, and as magneto-priming to improve seed quality [12].

The use of microwaves in green soybean samples proved to be efficient in the reduction of the drying time [13]. The technology also presented an increase in the antioxidant activity in bean fava [14]. *Phaseolus vulgaris* L. seeds germinated after exposure to the electromagnetic field obtained a bio-stimulus in the frequency of 890–915 MHz [15]. The bio-stimulation action on soybean seeds is little known at the level of the electromagnetic spectrum of microwaves, especially regarding its latent effect, i.e., the longevity of the seeds.

The microwaves, which are part of the electromagnetic spectrum, operate with frequencies ranging from 300 MHz to 300 GHZ, corresponding to the length of 1 m to 1 mm, respectively [16]. Therefore, there is a wide range of use this energy, which can be explored to obtain some bio-stimulation effect on soybean seeds. Thus, the magneto-priming performed by microwaves in 2.45 GHz frequency was approached in this study to verify the benefits promoted in soybean seeds after their exposure and their response after storage at a high temperature and high relative humidity conditions.

## 2. Results

The seed water content before the bio-stimulation process was 6.39% for the cultivar C01 and 10.24% for the C02 cultivar, and the water content reduction during the magneto-priming was 0.12 and 0.6% for cultivars C01 and C02, respectively. The process of reducing the water content in the seed is presented in Figure 1. The drying curves were adjusted by the Bleasdale model, whose parameters are shown in Table 1.

The reduction in seed water content occurred as expected, being more intense at the beginning of drying, and reducing in speed over time. In this study, the seeds already had a low water content; therefore, the reduction of this index was not significant.

The differences between soybean cultivars were evaluated from the germination pattern test (GPT). Figure 2a–c shows the total germination indexes, normal seedlings in the first GPT count, and abnormal seedling index, respectively.

From the analysis of the TPG, we can see that cultivars C01 and C02 have differentiated physiological quality. For the control group, cultivar C02 was superior to C01 in the overall germination index and the vigor test of the first count, and also presented a lower seedling index of abnormal plants. In the group subjected to magneto-priming, the highlight was cultivar C2, which presented a higher index of normal seedlings in the vigor test of the first count. The authors in [17] also identified an increase in normal seedling indices in the first count, and a reduction of abnormal seedlings in *Phaseolus vulgaris* L. seeds after microwave exposure for a period ranging from 15 to 120 s.

The analysis of the germination test showed that the groups submitted to magneto-priming were improved in at least one factor and were superior to the control group. Cultivar C01 had a significant difference in the overall germination index in relation to the control group. Cultivar C02, as already cited, was superior to the control group in the first count test.

For a specific performance evaluation between the group exposed to magneto-priming and the control group, the cultivars were analyzed separately. For this evaluation, the same germination test was analyzed to verify the differences within each cultivar before and after the magneto-priming.

Figure 3 shows the results of germination, first count, and abnormal seedlings, for cultivars C01 and C02 positioned on the left and on the right, respectively.

The analysis of each cultivate separately confirms the results presented by the Fisher’s LSD test, that is, cultivates C01 and C02 that received the biostimulation had superior performance to the samples that did not receive the treatment in the germination indexes and the first count, respectively. The data also corroborate with [18], who studied seeds of the species Gleditschia gleditsia, Caragana arborescens, Laburnum anagiroides, and Robinia pseudoacacia and verified the increase in germination indexes after exposure the microwave with powers of 255, 425, and 850 W over a period of 30 s, even indicating the breaking of dormancy of these species.

The test also demonstrates that the samples treated with the magneto-priming were more uniform than the control samples. That is, the dispersion around the value of the median was lower, leading to the formation of stronger and more uniform seedlings, as can be perceived by Figure 4.

Figure 5 presents part of the normal seedlings produced in the germination test for the C02 cultivar and illustrates the quality of the seedlings germinated after the magneto-priming. The difference in size and uniformity of seedlings can be perceived from the image.

Because cultivar C01 had a larger number of abnormal seedlings, it did not present visible aspects of this difference. However, by the graphical analysis of the results of the germination test, we saw that the cultivar was also benefited by the treatment, presenting better germination and uniformity parameters.

The data of the seedling length test were also analyzed for each cultivar individually. Figure 5 shows the results of the analyses performed in the samples of the seeds immediately after the treatment was applied.

Cultivar C02 showed a significant difference in the seedling length test. The indices evaluated were higher for the biostimulated seeds in their total length and in the root length of the seedlings. The fact that the treatment has enabled greater root development for seedlings is particularly relevant with regard to the water absorption capacity and nutrients that this plant can develop in the field.

The data obtained in this study corroborate with [19], who studied soybean seeds and also identified an increment in the germination indexes, seedling growth, and in the dry mass of seedlings germinated after microwave exposure for 20 s at a power of 200 W.

This cultivar had already demonstrated good performance in the germination test and the seedling length test, and this is a second confirmation that the electromagnetic energy has brought benefits in the physiological indices of this cultivar. Cultivar C01 had no significant difference in this test, however, and the medians presented in the graphs had a higher value in each of the three parameters evaluated. It is also noteworthy that the variation of the data obtained was slightly smaller in two of the three parameters evaluated.

The results the fresh mass and dry mass of the roots and hypocotyl is presented in Figure 6 below.

The fresh and dry mass of the hypocotyl of cultivar C01 showed significant differences for the biostimulated samples. Although the length was not different in relation to the control group, its weight indicates that this region may have formed a larger caliber structure, which would justify the increase of the fresh weight and dry weight of the sample.

Samples of normal and abnormal seedlings found in cultivar C01 of the control group are shown in Figure 7. During the analysis, it was noticed that the portion of abnormal seeds showed an almost predominant deformation in the formation of the root part. However, after further investigation, it can be inferred that such deformity induced the biostimulated seeds to develop more expressive growth of the hypocotyl and not of the root, as observed in cultivar C02.

The test of fresh mass and dry mass of cultivar C02 presented in Figure 8 presents significantly higher values for the seeds submitted to magneto-priming. This test confirms that not only the length of the roots was higher, but also its mass content. This result is more indicative that the magneto-priming was able promote a significant increase in the physiological parameters of the seedlings.

For the germination process to begin to occur in soybean seeds, it is necessary that a certain amount of water is absorbed. The water absorption in the seeds can bring relevant information on the quality of the seeds. Figure 9 shows the percentage data of water absorption of the seeds in cultivars C01 and C02, for the control groups, and those exposed to the MP, respectively. Table 2 shows the parameters of the models adjusted to the process of water absorption by the seeds.

The water absorption capacity of the biostimulated seeds was faster and higher, on average, than the seeds in the control group for cultivate C02. The authors in [20] report that this phenomenon may occur when the seeds of the group with a higher curve present osmotically more reactive compounds in their composition. In other words, this means that proteins and carbohydrates may have broken down, making amino acids and sugars available for the accelerated cell multiplication process. The germinability indexes shown in Table 3 demonstrate that for the C02 cultivate, there was a higher germination speed confirmed by the indices T50, T10, and TMG. In the C01 cultivate, no significant differences were observed in this test; however, the mean indices T50, T10, and TMG were lower, indicating a tendency in the improvement of these parameters.

Cultivate C02 continued to present a better performance in the vigor indices described in Table 3. The cultivar had a higher germination speed, very possibly because it presented more bioavailable energy in the form of amino acids and sugars in its composition. However, the AUC uniformity index was lower. It is known that the application of electromagnetic energy may not be homogeneous within the radiation incidence chamber. This was most likely the factor that brought the reduction of this index; however, for cultivate C01, the same behavior did not occur. Cultivate C01 also showed no difference in the other indices, which is indicative that the performance of the benefits of exposure to magneto-priming can be varied according to cultivate.

The soybean samples from both cultivates before and after magneto-priming were submitted to the longevity study, as previously described. The data on the germination probability during seed storage are shown in Figure 10 and Table 4. The longevity behavior presented a superior curve for the cultivates that were submitted to magneto-priming.

The P50 values calculated by the logit model confirm that the seeds’ physiological quality indices were higher for the seeds exposed to magneto-priming. According to the logit model, the treated seeds had an increment of the P50 of 0.88 and 0.95 days more than the untreated seeds for cultivars C01 and C02, respectively. This represents an increase of 11% of the cultivar C01 and 8% of the cultivar C02 in the average longevity time of these cultivars. The effect of the MP was also represented by the survival analysis given by Box–Cox. (Figure 11).

The analysis of regression of the Box–Cox had greater survival for the seeds treated with the MP. This result reveals that the benefits promoted by the MP occurred not only in the evaluation shortly after the priming procedure, but are also maintained during the seed life, even when the seeds were subjected to a high temperature and high relative humidity. This information is revealing because it can be inferred that MP was able to prolong the seed viability period, which can bring a number of benefits to commercial purposes and can be used for the preservation of species in seed banks.

Studying the electromagnetic biostimulation procedure of seeds harvested at the end of late maturation with the use of microwaves at a power of 0.2 W/g for 15 min at 40 °C, there was the promotion of drying the seeds and, at the same time, enhancing the quality of the physiological effect on treated seeds. The parameters of germinability, seedling length, water absorption, and fresh and dry mass improved after exposure to the drying process. Microwave drying was able to promote increases in the parameters of seed longevity. However, results may vary between different cultivars.

In the seeds that were submitted to treatment, the genetic expression related to germination and longevity was analyzed (Figure 12), and demonstrated a significant increase for seeds that were previously hydrated for 6 h when compared to seeds that were not subjected to the drying treatment (control), suggesting the activation of the processes associated with the development of RNA cell elongation meristem to seeds submitted to the magneto-priming.

The gene expression, shown in Figure 12, suggests that the seeds subjected to the microwave stimulation process are more prepared for transcriptional reactions related to the germination event, physiological quality, and longevity.

## 3. Discussion

The use of electromagnetic biostimulation in seeds has been the object of study in several studies. Soybean seeds had an increase in physical and chemical indices after biostimulation in pulsed electromagnetic field equipment at 10 Hz and 1500 NT for 20 days, 5 h a day [21]. However, the authors in [22] identified a reduction in soybean seedling growth after microwave exposure with a frequency of 900 MHz. In this study, it was evidenced that the increase of physical/physiological parameters can be safely acquired, even with a reduced time of exposure of the product to electromagnetic energy.

The results in [23] showed the effect of pre-treatment of a static magnetic field of 200 MT for 1 h in seeds subjected to UV-A and UV-B radiation stress and found lower rates of hydrogen peroxide and antioxidant enzyme activity. The longevity of the seeds had, on average, higher germination values during almost the entire storage period under a high temperature and high relative humidity conditions. This result is indicative that the biostimulus produced in this study has possibly promoted a similar effect, reducing the antioxidant activity in the seeds and providing greater longevity.

The use of a low-frequency electromagnetic field was tested in Moyashi bean seeds [24], which confirmed an increase in the calcium and phosphorus indices and in the protein content, but were not observed incrementally in seed length and germination. In this study, the germination index of C01 also obtained a significant increase, even though it is already high, and although this cultivate did not present an increase in the length and speed of germination, a higher fresh weight was identified and in the dry seedlings, better longevity performance was observed, which is indicative that biochemical changes may have occurred in the seeds.

The authors in [25] studied sunflower seeds exposed to an electromagnetic field with 5.48 MHz radiofrequency and identified that the treatment altered the phytohormones balance in the seedlings and the expression of long-term genes in the leaves, which led to the expression of proteins involved in photosynthetic processes. Therefore, judging by the increase in the longevity of soybean seeds observed in this study, it can be inferred that a similar process has occurred. However, this analysis lacks further investigation.

The results presented in this study show that magneto-priming with microwaves can bring benefits to the physiological quality of soybean seeds and can even be used during the drying process of the seeds. The benefits of increasing seed quality have the potential to be used as a pre-treatment for other types of priming that usually reduce seed longevity of high vigor seeds [26]. However, microwave exposure technology still lacks advances in engineering that enable its use on a large scale.

Currently, the industry has advanced in the development of equipment that allows the use of electromagnetic energy in the microwave spectrum for the drying of agricultural and pharmaceutical products. In this study, it is evident that electromagnetic energy has the potential to be used in the drying processes of agricultural products with the benefit of bringing improvements in their physiological characteristics. The use of microwaves has been little applied to improve the characteristics of agricultural products, especially due to the need to adjust existing equipment in the dosage and stability of energy application, making it safe for its use on a large scale. This study demonstrated that there are great possibilities of gains with equipment improvement that use this technique, and we suggest further investigations in this area, especially using low doses.

## 4. Materials and Methods

### 4.1. Preparation of Samples

Two soybean varieties were selected for this study. The seeds were produced in 2018 in Montreal, Quebec, Canada. The varieties acquired in the local trade were encoded in this article as C01 and C02, the samples are from different cultivars and have different percentages of vigor. After collecting the samples, both were conducted to the post-harvest laboratory where they remained stored at ambient conditions at 21 °C and 55% RH, up to the time of each analysis.

### 4.2. Electromagnetic Biostimulation

The seeds submitted to biostimulation were separated in samples of 100 g. Each sample was placed in a bracket inside an electromagnetic energy generator that operated at a frequency of 2.45 GHz with a wavelength of 11 cm. During the biostimulation, the seeds received a power of 0.2 W/g ± 0.1 at intervals of 30 s for 15 min. The equipment used is a microwave emitter that has been adapted to produce electromagnetic energy in the power and conditions of the study. The procedure was carried out with a current airflow at a maximum temperature of 40 °C, which allowed the removal of water from the product, also promoting its drying (Figure 13—Source: [27]).

The water content of the product was measured before and after biostimulation by the oven method in three replications, at 105 °C ± 1, during 24 h [28]. A water content reduction curve was adjusted using the Curve Expert software 2.6.

### 4.3. Analysis of the Germinability and Vigor of Seeds

To perform the germination pattern test (GPT), the seeds were placed on two sheets of paper towel moistened with water in the amount of 2.5 times their own weight. Then the seeds were covered with one more sheet of moistened paper towel and wrapped in a cylindrical shape. The paper rolls were assembled with six replications of 50 seeds and placed in plastic bags, properly sealed, and placed in a germination chamber at 25 °C for 8 days. At the end of 5 days in germination, the normal seedlings produced were counted, and the percent represented the first count vigor test. After 8 days of germination, all the seeds that emitted at least 1 mm of root protrusion were considered germinated. 

The seeds were classified as normal seedlings—germinated seeds that had root, hypocotyl, and leaves with normal development–and abnormal seedlings, those that did not have at least one of these three parts, or had some physical damage, as described in the International Rules for Seed Analysis [28].

The seedling length test is a vigor test, which was performed with six replications of 10 seeds each. The seeds were lined in the upper third of two sheets of paper towel moistened with water, at 2.5 times their weight. The seeds were covered with a sheet of paper towel moistened and curled in cylindrical shape. The six rolls of each treatment were placed in a plastic bag, sealed, and placed in a germination chamber for eight days at 25 °C. At the end of eight days, the seeds that produced normal seedlings had their length measurements referring to the root and hypocotyl measured [28].

The analysis of the fresh mass of seeds germinated in the seedling length test was performed with the measurement of the fresh weight of the root and hypocotyl, separately. For the dry mass analysis, the samples of the fresh mass were placed in an oven at 80 °C for 24 h. After this period, the samples were placed in a desiccator for thirty minutes and weighed on scales with a precision of 0.0001 g [28].

The water uptake curve was performed with the measurement of the individual weight of 30 seeds, arranged in a Petri dish, on two sheets of paper towels. Each seed had its weight previously measured on a 0.0001 g precision scale. In each Petri dish, 20 mL of distilled water was added. During the test, the seeds were kept in germination chambers at 25 °C, during a period of 8 days. However, the weight of each seed was again measured at intervals varying from 3 to 24 h during the incubation period. During each weighing, each seed was placed on absorbent paper, certifying that there was no excess of free water around the tegument. At the end of the test, the water absorption curve versus time was constructed from the percentage values of the absorbed water weight. The adjustment of the water uptake curve models was performed using the software Curve Expert 2.6.

From the germination data obtained through the water uptake curve test, the germinability indexes GMAX, t50, U8416, t10, AUC, and TMG were calculated by GERMINATOR software [29]. The GMAX index represents the maximum germination obtained in the T50 test, which is the time required to reach 50% of germination. The index of uniformity U8416 represents the time until germination occurred between 16 to 84% of the examined lot. The germination parameter T10 is equivalent to the time required for 10% of the seeds germinate. The AUC index represents the integration of the adjusted curve between t = 0 and a final t defined as the time of the last reading performed in the test, and the TMG index is the average time passed until germination of the sample lot occurred.

### 4.4. Seed Longevity Analysis

The study of seed longevity was accomplished by placing the stored seeds under conditions of thermal stress and high relative humidity. For this purpose, the seeds were placed in airtight plastic pots with 100 mL of a saturated saline solution, produced with NaCl, which simulated the condition of 75% RH inside the pots. Plastic screens were adjusted in the solution within each pot, which allowed the seeds to be suspended, thus not touching the solution. The pots with the seeds treated and with the control samples were placed in an oven at a constant temperature of 42 °C ± 1 [30].

After the longevity test was performed, the germination test was periodically performed to identify the number of viable seeds in the lot. The germination test was performed until 100% of seed viability was verified. After the test was performed, a curve was adjusted, demonstrating the viability of the seeds during the storage period. 

The longevity modeling was performed from the logit link function, whose adjustment equation is given by F (x) = ln (x/(1 − X), where x is the percentage of germination, given in decimal. The line formed by the logit scale versus time allows us to calculate the P50. The P50 is the index that indicates the period, in days, in which the seed lot germination reaches 50% of its viability. This index is obtained after the transformation of the germination data into the logit scale. The logit scale is linear, therefore, the line produced by the model allows us to find the P50, according to Equations (1) and (2).
Y = β − αX,(1)
XP50 = β/α,(2)
in which β is the intercept, that is, the initial viability of the lot; α is the slope of the line and indicates the velocity at which the decrease in viability occurs; X is the period (days); and XP50 represents the number of days in which the loss of 50% of the viability of the lot occurred. The survival analysis was also performed by the Box–Cox regression.

### 4.5. Molecular Analysis—Quantification of the Expression by Real-Time q-PCR

Cultivar C02 was selected for molecular analysis because it was the sample that best represented the performance of the magneto-priming. Therefore, the embryonic axes of the control and treated samples were frozen in liquid nitrogen after 10 h of soaking in water at 25 °C. The total RNA was extracted using the NucleoSpin^®^ RNA Plant kit (Macherey-Nagel, Düren, Germany) according to the manufacturer instructions. Three biological replicates of seeds were used. For cDNA synthesis, the High-Capacity cDNA Reverse Transcription kit (Applied Biosystems, Victoria, Australia) was used.

We previously selected target genes related to longevity and germination using the literature to perform the analysis. Based on the results of the literature analysis, we selected: HSFA3, HSP21, HSP17.6b [31], EXPA10 [32], and ABI3 [33]. The sequence of primers used are listed in Table 5. The amplification was performed in real-time q-PCR using the KiCqStart^®^ SYBR^®^ Green qPCR ReadyMix kit (Sigma-Aldrich, St. Louis, MO, USA), using an Eco Real-Time optical thermal cycler (Illumina, San Diego, CA, USA). The data obtained were analyzed using Illumina’s EcoStudy v5.0 software. The expression was calculated by the 2^−∆∆Ct^ method [34], using two reference genes, Importin beta-2 subunit family protein (Glyma.20G106300) and 20S proteasome subunit beta (Glyma.06G078500) [35].

### 4.6. Statistical Analysis

The seeds of the cultivars chosen for the study were separated into two groups: the control group, which did not receive magneto-priming, and the group of seeds that were treated, i.e., that were exposed to magneto-priming, encoded as MP. Firstly, the two soybean cultivars were examined for their germination capacity by Fisher’s LSD test at the level of significance of 0.05. To evaluate the effect of magneto-priming within the same cultivar, the non-parametric Mann–Whitney U test was applied with a significance level of 0.05. The software R 3.6.1 was used to perform the analyses.

## 5. Conclusions

The procedure of electromagnetic biostimulation of seeds using microwaves at the power of 0.2 W/g during 15 min to 40 °C was able to promote the drying of seeds and, at the same time, improve their physiological quality. The parameters of germinability, seedling length, water absorption, and fresh and dry mass obtained some type of improvement after exposure to magneto-priming.The magneto-priming from the microwave was able to promote increments in the parameters of seed longevity.The genes involved in seed germination and longevity showed superior expression after microwave exposure.

## Figures and Tables

**Figure 1 plants-12-01477-f001:**
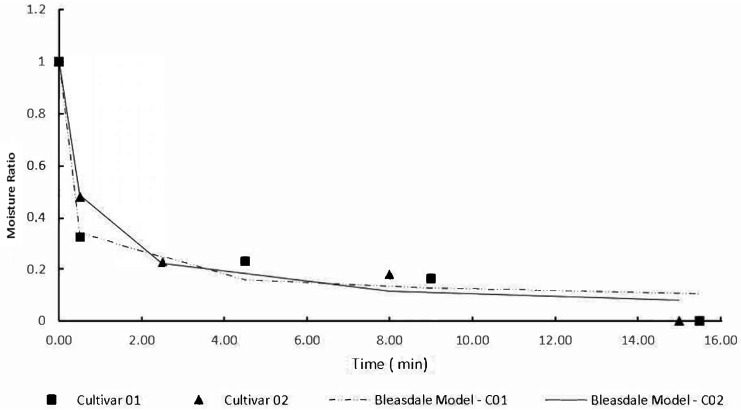
Reduction of water content during the magneto-priming.

**Figure 2 plants-12-01477-f002:**
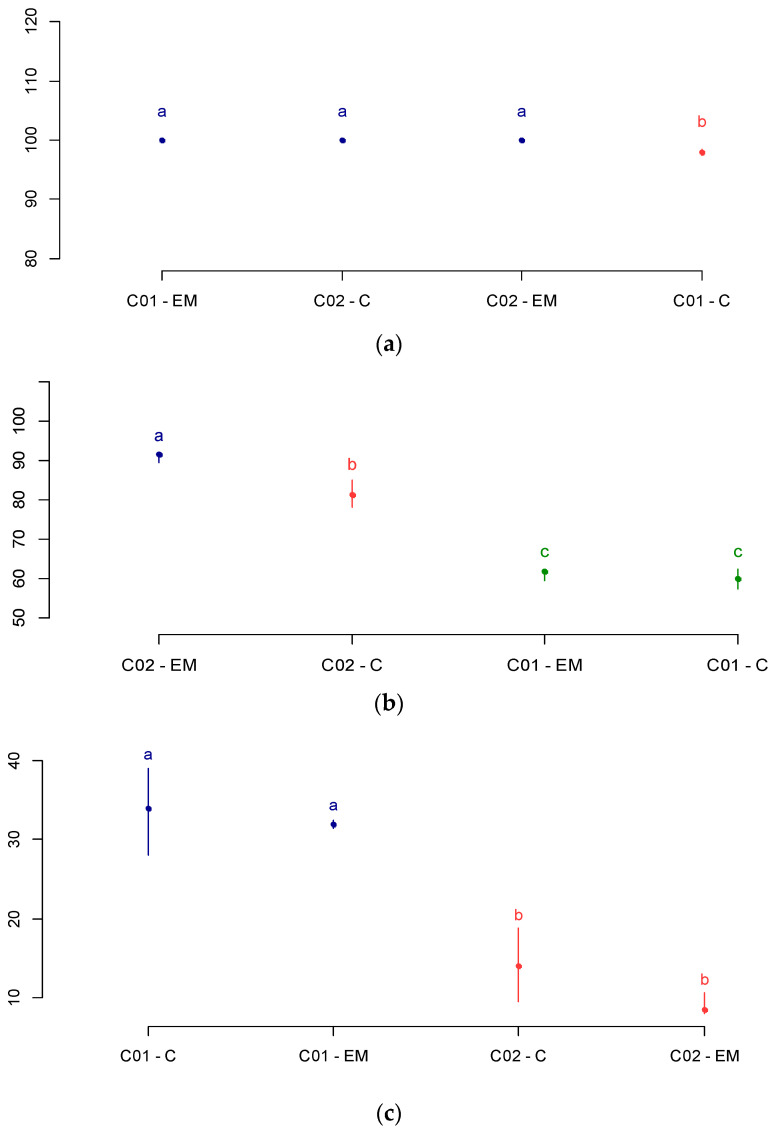
Germination pattern test (GPT). (**a**) Total germination; (**b**) normal seedlings in the first count test; (**c**) abnormal seedlings. C—control group; EM—electromagnetic bio stimulation (magneto-priming). Average followed by the same later end color do not differ significantly by the Fisher’s LSD test.

**Figure 3 plants-12-01477-f003:**
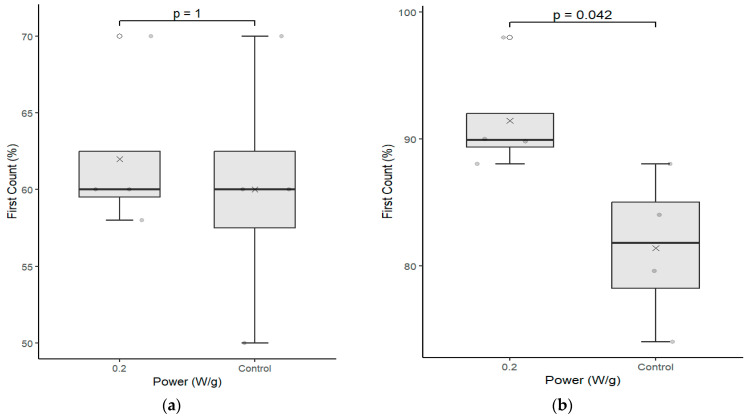
Germination pattern test between the control group seeds and those that underwent magneto-priming (0.2 W/g). The left, figures (**a**,**c**,**e**), are related to cultivar C01; the right, figures (**b**,**d**,**f**), refer to cultivar C02. The differences were estimated by the Mann–Whitney test at a significance level of 0.05. The media, data, and *p* value are represented in the figures for the x, dots, and letter *p*, respectively.

**Figure 4 plants-12-01477-f004:**
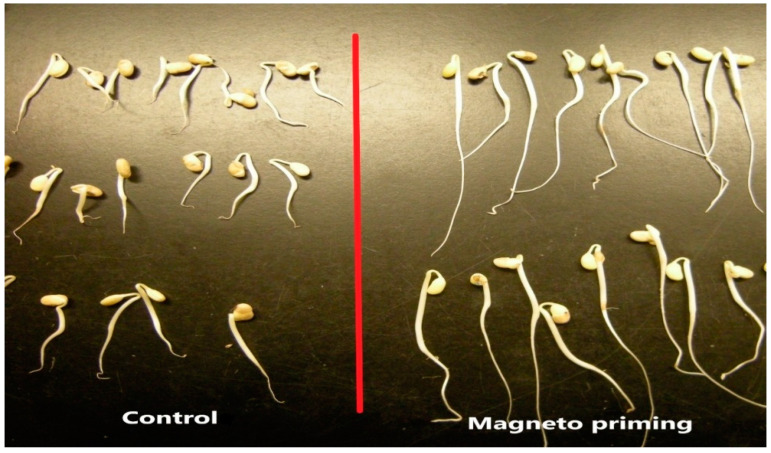
Seedlings originating from the germination pattern test for cultivar C02 before and after the magneto-priming.

**Figure 5 plants-12-01477-f005:**
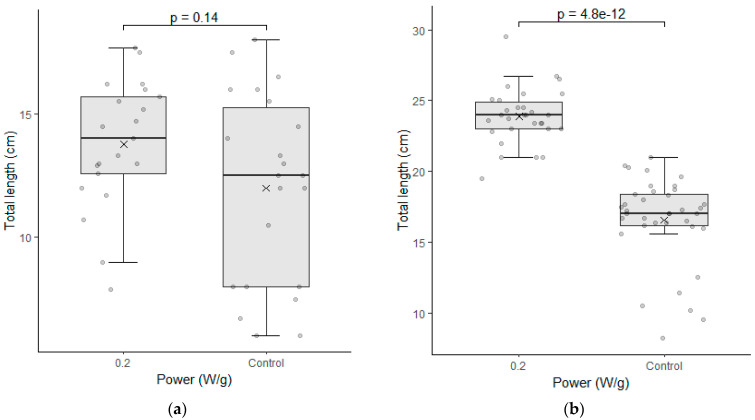
Seedling length test between the seeds of the control group (0 W/g) and those subjected to the magneto-priming (0.2 W/g). On the left, figures (**a**,**c**,**e**) are related to cultivar C01; on the right, figures (**b**,**d**,**f**) refer to cultivar C02. The differences were estimated by the Mann–Whitney U test at a significance level of 0.05. The media, data, and *p* value are represented in the figures for the x, dots, and letter *p*, respectively.

**Figure 6 plants-12-01477-f006:**
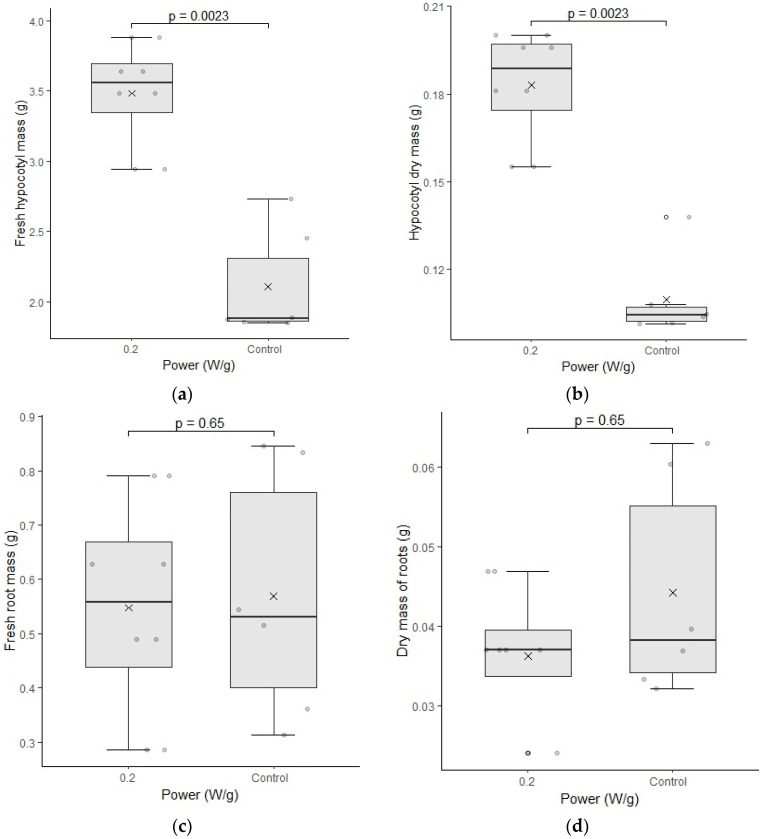
(**a**–**d**) represent the test of fresh mass and dry mass of cultivar C01. The differences were analyzed using the Mann–Whitney U test at a significance level of 0.05. The media, data, and *p* value are represented in the figures for the x, dots, and letter *p*, respectively.

**Figure 7 plants-12-01477-f007:**
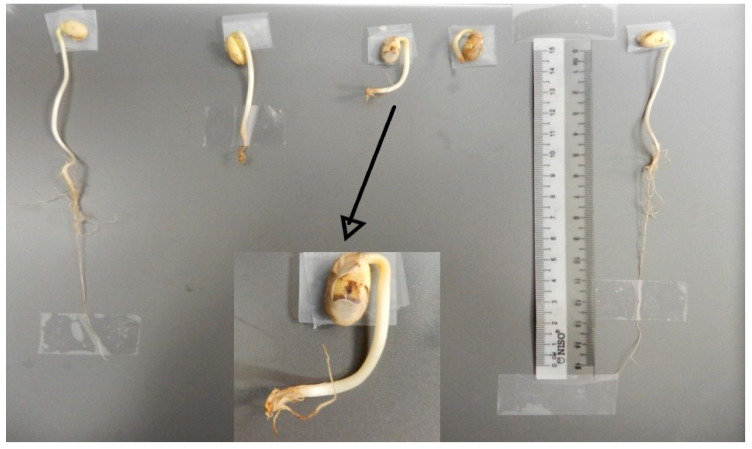
Normal (extremes) and abnormal (center) seedling samples produced by cultivar C01 in the control group.

**Figure 8 plants-12-01477-f008:**
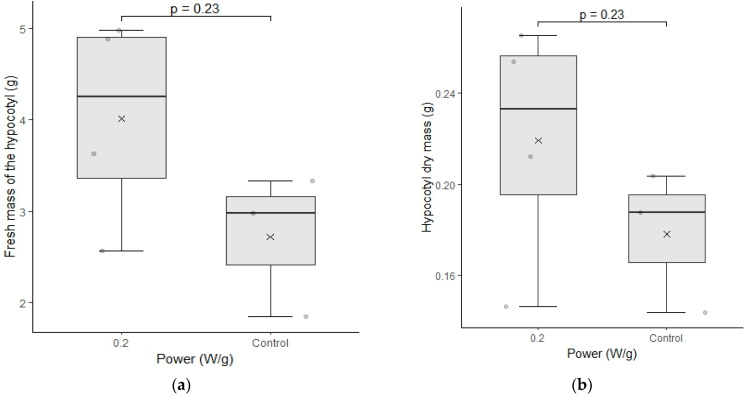
(**a**–**d**) represent the test of fresh mass and dry mass of cultivar C02. The differences were analyzed using the Mann–Whitney U test at a significance level of 0.05. The media, data, and *p* value are represented in the figures for the x, dots, and letter *p*, respectively.

**Figure 9 plants-12-01477-f009:**
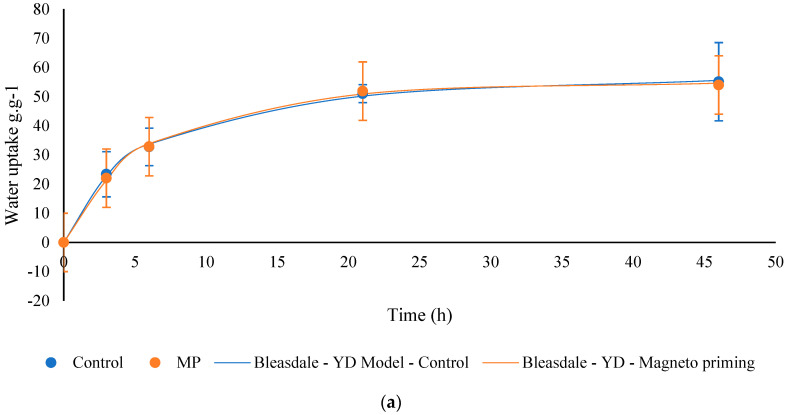
Water absorption curve of soybean seeds before and after exposure to magneto-priming. (**a**) Cultivar C01; (**b**) cultivar C02. The error bar indicates the standard deviation (*n* = 3).

**Figure 10 plants-12-01477-f010:**
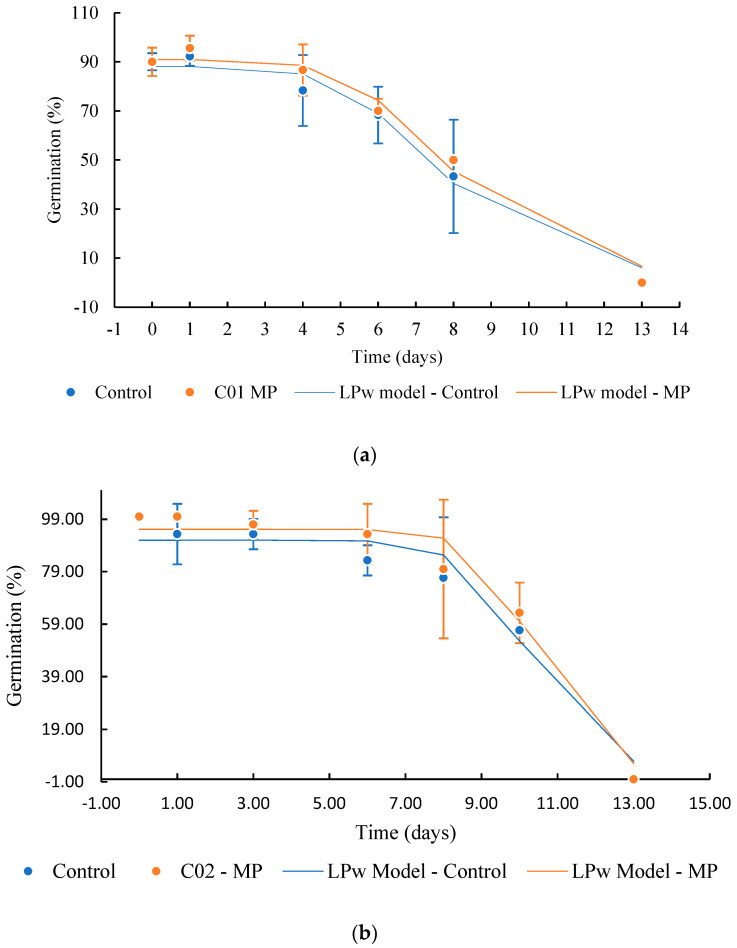
Survival curve of soybean seeds stored at the temperature of 42 °C ± 1 and 75% RH. (**a**) Cultivate C01, (**b**) cultivate C02. The logistic power model (LPw) was used to adjust the curve behavior. The error bar indicates the standard deviation (*n* = 3).

**Figure 11 plants-12-01477-f011:**
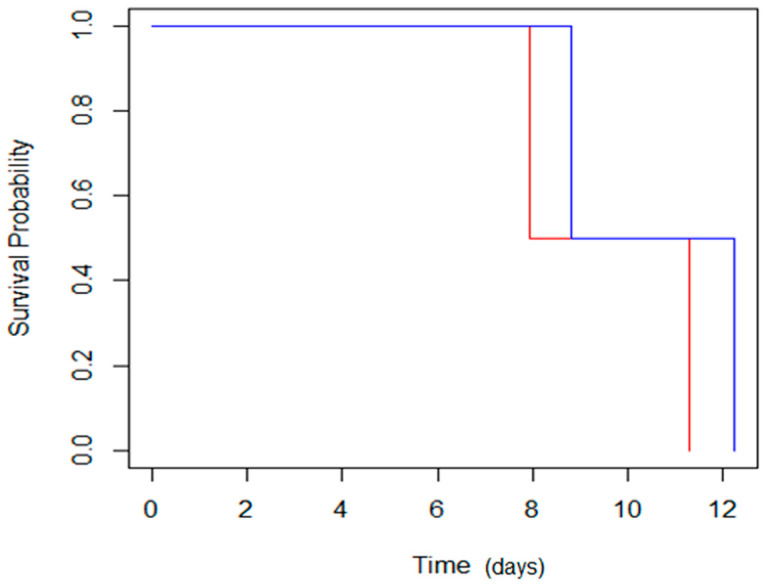
Regression of Box–Cox by the Breslow method; the blue line indicates survival of the seeds treated with magneto-priming, and the red line represents the untreated seeds. The values were obtained by the Box–Cox regression model, with the likelihood ratio test = 0.62 and *p* = 0.4, and a 0.95 confidence interval.

**Figure 12 plants-12-01477-f012:**
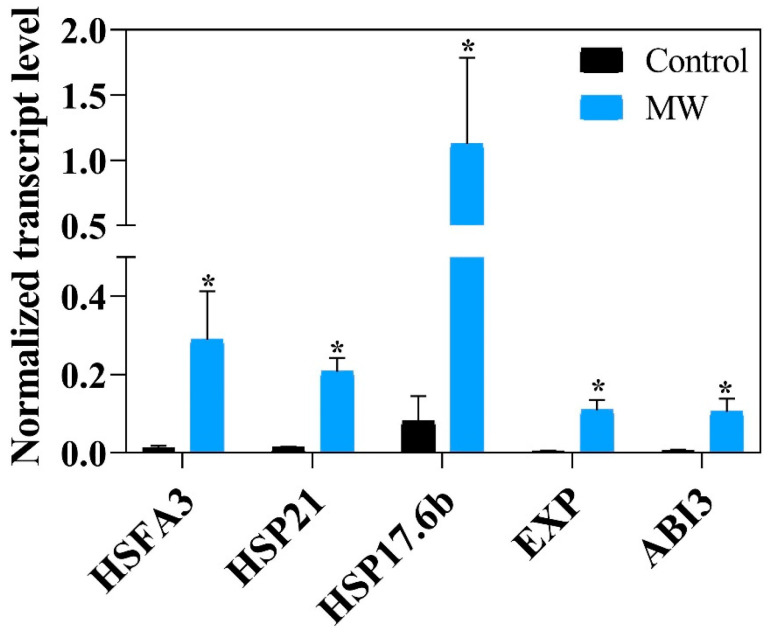
Normalized transcript level at 6 h after imbibition on soybean seeds of the HFA3, HSP21, HSP17.6b, EXP, and ABI3. Each bar represents the means from two samples of 15 embryony axis. The symbol (*) shows a significant difference between the MW and control treatment by *t* test at a 0.05-confidence level.

**Figure 13 plants-12-01477-f013:**
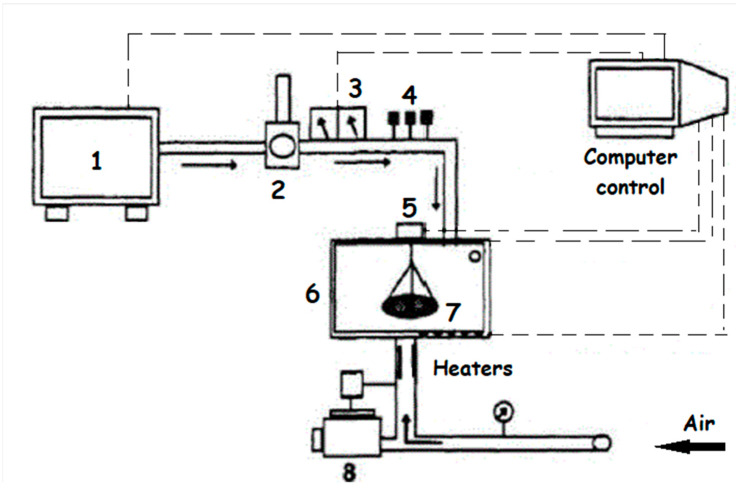
Dryer equipment (1: microwave generator, 2: circulator, 3: power meters, 4: adjusting controls, 5: weight balance, 6: microwave chamber, 7: sample support, and 8: fan).

**Table 1 plants-12-01477-t001:** The Bleasdale model parameters adjusted for the reduction of the water content in soybean seeds treated with the magneto-priming.

Cultivate	Temperature °C	Power W/g	a	b	c	R^2^ *
C01	40	0.2	1.0005	42.777	2.8802	97
C02	40	0.2	1.0014	4.7325	1.6885	98

* Significance level of 0.05.

**Table 2 plants-12-01477-t002:** Parameters of the Bleasdale YD model given by y = x (A + bx^θ^)^−1/θ^ and adjusted for the water absorption curve in soybean seeds before and after exposure to magneto-priming (MP).

Cultivar	Treatment	a	b	I	R^2^ *
C01	Control	7.02 × 10^−2^	1.07 × 10^−2^	1.10	0.99
	MP	3.71 × 10^−2^	1.90 × 10^−3^	1.55	0.99
C02	Control	5.63 × 10^−12^	5.91 × 10^−12^	10.53	0.99
	MP	1.39 × 10^−11^	1.76 × 10^−17^	8.08	0.99

* Significance level of 0.05.

**Table 3 plants-12-01477-t003:** Germinability indexes obtained through the water absorption test for the seeds before and after the magneto-priming (MP) process.

Cultivate	Treatment	G_MAX_ (%)	T_50_ (h)	U_8416_ (h)	T_10_ (h)	AUC	TMG
C01	Control	100 a *	23.9 a	15.2 a	15.8 a	22.9 a	24.2 a
	MP	100 a	20.5 a	15.7 a	12.5 a	25.8 a	21.1 a
C02	Control	100 a	30.5 b	11.6 a	23.9 b	16.0 a	30.3 b
	MP	100 a	22.9 a	21.7 a	12.6 a	22.3 b	21.7 a

* Treatments that share the same letter do not differ by the test and t at the level of 0.05 significance.

**Table 4 plants-12-01477-t004:** Models for adjusting the longevity curve of soybean seeds before and after exposure to magneto-priming (MP).

Test	Cultivate	Model	Model’s Parameters							
			Control				MP			
			a	b	c	R^2^ *	a	b	c	R^2^ *
Germination	C01	LPw	88.11	7.74	5.03	0.89	90.98	7.99	5.21	0.96
	C02	LPw	90.99	10.29	10.78	0.96	95.17	10.44	12.38	0.97
Longevity prediction			B	P	P50	R^2^	B	P	P50	R^2^
	C01	Logit	2.643	0.332	7.95	0.66	2.804	0.317	8.83	0.68
	C02	Logit	3.88	0.343	11.30	0.34	2.814	0.229	12.25	0.37

LPw—Logistic power model given by y = a/(1 + (X/b)^c^); logit model given by: y = β − αx and P50—time period of 50% of seed survival, given by X_P50_ = β/α. * Significance level of 0.05.

**Table 5 plants-12-01477-t005:** Primer sequences used as target and reference genes (mRNAs) in the RT qPCR reactions.

Gene	Forward (5′–3′)	Reverse (5′–3′)
20S proteasome subunit *	CACCAACACACGATACAACT	TCCCAACCACCAACAATTAACC
Importin beta-2 subunit family protein	GATAATAAGCGGGTCCAAGAG	GTCATCAGGTGCTTCAGTATAA
HSP 21	AACATGCTGGTGGTGAAG	AGGGCTATCCTGTGGTTAT
HSFA3	CATCAGGTTGGTGGCAATA	GCATTAGCACACTCCTTTCT
HSP17.6b	TGCGGATGTGAAGGAATATC	AAGCACGTTGTCGTCTTC
EXP	TTCGCATTGCACAATACAGAG	TTATGAGGACCAAGTTAAAGTAGG
ABI3	GCCATACCATCACCAACAA	CGAACTCGAACTAGAACTGC

* Reference gene.

## Data Availability

Not applicable.

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
