# Peer review of "The Effect of Magneto-Priming on the Physiological Quality of Soybean Seeds"

_plants, 2023, doi:10.3390/plants12071477_

Round 1
Reviewer 1 Report
Manuscript ID: plants-2193366
In the manuscript titled as “THE EFFECT OF MAGNETO PRIMING ON THE QUALITY AND LONGEVITY OF SOYBEAN SEEDS EXPRESSING GENES HSFA3, HSP21, HSP17.6b, ARF, EXP, and ABI3” by Rute Quelvia de Faria et al., the authors used microwave apparatus on two different soybean cultivar seeds and tested their responses in terms of seeds germination and seedling growth traits. The manuscript included detailed observation of the physiology of seeds and seedlings after microwave treatment.
The biggest problem I found in this manuscript is about the gene expression part. There are several questions I have for the authors:
1. How were the differentially expressed genes found? The authors only mentioned about real time PCR in the methods part, so I assume the authors only tested a short list of candidate genes that they are interested with, based on previous literature? Then how do you know other genes are not differentially expressed in this seeds? If the authors did the DEG analysis in a different way, please describe clearly in the materials and methods.
2. In the methods part, the authors mentioned that they used DESeq2 for DEG analysis, however, I don’t see why you would need to use DESeq2 for RT-PCR results, please further explain.
3. Based on the questions about these DEGs, I don’t think it is a good idea to bring readers attention to these genes too much by mentioning them in the title. Actually, the grammar of the title seems wrong. I would suggest removing these gene names in the title.
4. If the authors would still keep the DEGs in the results, which is good, then they would also need to make some more instructions about these genes either in the introduction or in the discussion.
Another major issue is about the citation style. Please make sure that both the in-text citation and the reference list meets the citation style of Plants.
Moreover, I would suggest the authors work on the introduction part a bit more.
Small problems:
There is a lot of extra space in the text.
Figure 5 is stretched out of shape, please pay attention to the width/heigh ratio of the original file.
Author Response
Dear Reviewer,
Regarding the notes described in your review, we inform you that:
1.
An erroneous note was made in the methodology, which led to a misinterpretation. The text has already been corrected, and we inform you that genes already reported in the literature were used, and this was clarified in the corrected text.
2. DEG analysis was not performed, as mentioned earlier, and the text has been corrected.
3. We agree with the revision and the title has been readjusted.
Thank you for your notes and suggestions.
Reviewer 2 Report
It seems to be an interesting study. The experiments were nicely designed and executed,. Ms is also well written. However, the data should be repeated in some other crops also, which will make the study more useful for other people.
Figure quality needs to be improved.
In case of RT PCR experiments, I could not see any internal control gene. A minimum of two internal genes are recommended now a days for validation. Further, Authors showed upregulated expression of ARF after magneto priming, that seems to be artefact, because ARF has been earlier used as internal control in several studies , where it showed consistent expression even after stress treatments.
Therefore, I would suggest to recheck the data, and repeat the experiments as suggested.
Author Response
Dear reviewer
We agree with the reviewer's statement, and we want to clarify that the two reference genes were used, and now this is described in the text.
We also inform that due to a lack of literature that corroborates our data, the ARF gene was withdrawn.
Thank you for your notes and suggestions.
Round 2
Reviewer 1 Report
I’m glad to see that the manuscript has improved and the authors corrected their errors in the methods, and they have addressed most of my concerns from last round of review. The manuscript looks better now with some small issues:
Figure 11, there is no unit in the “Time” in the x-axis.
Figure 13, the resolution of the figure is too low.
The reference style is not consistent.
Author Response
Dear reviewer,
Your suggestions were very suitable, and we accepted them all. We changed figure 11, adding the unit for the x-axis, and in figure 13, we made a resolution fix.
We thank you in advance for your contributions.
Reviewer 2 Report
I could not see any correction highlighted in the revised Ms.
As per the response, authors did not find example of ARF. Here is one https://www.mdpi.com/2075-1729/12/7/941 , and there are many.
Highlight the correction in Ms then it is possible to see the revision, otherwise, I would have several other comments.
Author Response
Dear Reviewer,
We apologize for the mistake of not highlighting the changes made.
The file now contains all the previous highlights and those suggested in this new round of review.
We also apologize for the justification given for removing the ARF gene.
We appreciate your reference suggestion, it will certainly help us with this analysis. Despite this, the authors decided to keep it out of this study, as its analysis and discussion need more attention and time.
Thank you in advance for your attention, and I apologize once again for this inconvenience.
Round 3
Reviewer 2 Report
It may be now accepted.